# *Ortho*-Phosphinoarenesulfonamide-Mediated Staudinger Reduction of Aryl and Alkyl Azides

**DOI:** 10.3390/molecules27175707

**Published:** 2022-09-05

**Authors:** Xingzhuo Li, Zhenguo Wang, Wenjun Luo, Zixu Wang, Keshu Yin, Le Li

**Affiliations:** 1PCFM Lab and GDHPRC Lab, School of Chemistry, Sun Yat-sen University, Guangzhou 510275, China; 2Center for Nephrology and Clinical Metabolomics and Division of Nephrology and Rheumatology, Shanghai Tenth People’s Hospital, School of Medicine, Tongji University, Shanghai 200072, China

**Keywords:** Staudinger reaction, organic azide, *aza*-ylide, 1,2,3-thiazaphosphole, *ortho*-phosphinoarenesulfonamide

## Abstract

Conventional Staudinger reductions of organic azides are sluggish with aryl or bulky aliphatic azides. In addition, Staudinger reduction usually requires a large excess of water to promote the decomposition of the *aza*-ylide intermediate into phosphine oxide and amine products. To overcome the challenges above, we designed a novel triaryl phosphine reagent **2c** with an *ortho*-SO_2_NH_2_ substituent. Herein, we report that such phosphine reagents are able to mediate the Staudinger reduction of both aryl and alkyl azides in either anhydrous or wet solvents. Good to excellent yields were obtained in all cases (even at a diluted concentration of 0.01 M). The formation of B-TAP, a cyclic *aza*-ylide, instead of phosphine oxide, eliminates the requirement of water in the Staudinger reduction. In addition, computational studies disclose that the intramolecular protonation of the *aza*-ylide by the *ortho*-SO_2_NH_2_ group is kinetically favorable and responsible for the acceleration of Staudinger reduction of the aryl azides.

## 1. Introduction

Staudinger reduction [1,2,3] is one of the most common transformations used to prepare amino compounds in organic synthesis. Recently, the biorthogonal property [4,5,6,7,8,9,10,11] of organic azides [12] has significantly enhanced the visibility of the Staudinger reaction in the context of chemical biology and medicinal chemistry. The classic Staudinger protocol employs trivalent phosphines, predominately triphenylphosphine, to mediate the reduction of organic azides under aqueous conditions via an *aza*-ylide (or iminophosphorane) intermediate [13,14] (Figure 1A). Although thermodynamically the formation of triphenylphosphine oxide provides a sufficient driving force for the Staudinger reduction, reactions are sluggish when the iminophosphorane intermediates are kinetically stable [15,16,17]. For example, iminophosphorane intermediates formed from aryl azides can be quite stable under neutral conditions, thus requiring either acids or bases to mediate their hydrolysis [18,19]. In addition, the amount of water in the system has a significant impact on the rate of Staudinger reduction, particularly for highly hydrophobic substrates. Recently, Ito, Abe, and coworkers [20] reported that *o*-(diphenylphosphino) benzamide accelerated the reduction of aryl and alkyl azides towards the corresponding amines (Figure 1B). In their report, aqueous conditions were still required for the reduction of aromatic azides. The authors indicated that a significant amount of by-product was formed without the addition of water. It therefore remains highly desirable to design novel phosphorus reagents that are able to mediate a broad-spectrum Staudinger reduction without the restriction of an aqueous solvent. Very recently, our group prepared a novel class of cyclic iminophosphoranes, namely (benzo[*d*])-1,2,3-thiazaphosphole (B-TAP) [21]. The improved stability of the B-TAP heterocycle towards hydrolysis suggests that the stronger P=N of B-TAP bond may drive a Staudinger reduction as the P=O of phosphine oxide did. Herein, we report the use of *ortho*-phosphinoarenesulfonamides as novel reagents to mediate the Staudinger reduction of both aryl and alkyl azides under either anhydrous or aqueous conditions (Figure 1C).

## 2. Results and Discussions

The B-TAP heterocycle was originally synthesized in our laboratory by the annulation of *ortho*-phosphinoarenesulfonyl fluorides with commercially available trimethylsilyl azide [21]. Further studies indicated that B-TAP can also be prepared by the oxidative condensation of *ortho*-phosphinoarenesulfonamide. We speculated that the formation of B-TAP could be achieved by the use of an organic azide as an oxidant. This hypothesis led us to explore *ortho*-phosphinoarenesulfonamide-mediated Staudinger reductions. We proposed that the introduction of an acidic sulfonamide group at the *ortho* position would facilitate the decomposition step of the original *aza*-ylide and thus the addition of water would not be necessary. In this way, the formation of a B-TAP heterocycle with a more stable P=N bond would provide the driving force for Staudinger reduction.

We selected methyl 4-azidobenzoate (**1a**) as a model substrate (Table 1). As expected, when **2a** was applied the major product was the triphenylphosphine-derived *aza*-ylide while the characteristic signals of methyl 4-aminobenzoate (**3a**) were not observed in the ^1^H NMR (entry 1, Table 1). Interestingly, *ortho*-phosphinoarenesulfonamide (**2b** and **2c**)-mediated reductions were both complete within 3 h. Reductions at lower concentration (0.01 M) with shorter reaction times indicated that **2c** gave faster rates than **2b** (entries 4–5, Table 1) and therefore **2c** was chosen for further investigation. The **2c**-mediated Staudinger reduction was found compatible with a broad range of organic solvents (entries 8–12, Table 1). The rate of **2c**-mediated Staudinger reduction in aqueous solvent (entry 13, Table 1) was similar to anhydrous solvents. This suggested that the addition of water did not affect the Staudinger reduction. This is in sharp contrast to other known staudinger reduction processes where water accelerates the hydrolysis step. Notably, the majority of B-TAP 4c (>90%) survived in aqueous THF and was not hydrolyzed to the corresponding phosphine oxide.

Next, we explored the substrate scope of the **2c**-mediated Staudinger reduction (Figure 2). Applying our standardized conditions, all aromatic azides evaluated gave excellent yields (**1a**–**1m**, Figure 2). Reductions of aliphatic azides were relatively slow at 30 °C and required elevated temperatures and longer reaction times for reaction completion, likely attributed to the decreased electrophilicity of alkyl azides. Gratifyingly, the reduction of aliphatic substrates **1n**–**1v** proceeded smoothly under the modified conditions, with satisfactory yields being obtained in all cases (Figure 2). Indeed, **2c**-mediated Staudinger reduction tolerated a broad range of functional groups and various substitution patterns. For instance, substrates with electron-withdrawing groups (**1a**–**1c**, **1h**, and **1i**), electron-donating groups (**1d** and **1f**) or reactive functionalities (**1e**, **1g**, and **1j**) are all fully compatible with the current protocol. Moreover, sterically-encumbered amines such as **3k**, **3l**, **3n**, and **3o** were obtained in high yields when *ortho*-substituted or branched substrates were used.

To demonstrate the synthetic potential of this methodology in a more complex setting, we explored the synthesis of a highly functional and sterically-hindered cinchonine-derived amine (**3w**). Staudinger reductions of cinchona alkaloid-derived azides have been reported to be problematic in some cases [22]. Their corresponding *aza*-ylide intermediates are reluctant to hydrolyze and are even able to be isolated by flash column chromatography. We proposed that accelerated hydrolysis could occur for **2c**-mediated Staudinger reduction since the *ortho*-SO_2_NH_2_ may facilitate the decomposition of the *aza*-ylide through intramolecular protonation. To our delight, the above proposal was verified by experiments with **2c** promoting an efficient Staudinger reduction of azide **1w.** The reduction was conducted on a 5.00-mmol scale and 1.21 g of product **3w** was isolated in 84% yield (Figure 3).

To elucidate the mechanism of the **2c**-mediated Staudinger reduction, we examined both the **2c**- and PPh_3_-mediated Staudinger reductions by DFT calculation (Figure 1) [23]. Both reductions share a three-step reaction pathway. In the initial step, the nucleophilic attack of either of the phosphine reagents on the azide generates a betaine intermediate. Then, this intermediate loses one equivalent of nitrogen gas to form an *aza*-ylide intermediate. Finally, the decomposition of the *aza*-ylide yields the amine product. In the conventional PPh_3_-mediated Staudinger reduction, the rate-determining step was the step from **Int-2′** to **TS-3′**. The high activation barrier (33.4 kcal/mol) was consistent with the fact that **Int-2′** was resistant to hydrolysis. In the **2c**-mediated Staudinger reduction, the rate-determining step was the step from **Int-1** to **TS-2** with a moderate activation barrier (24.7 kcal/mol). In addition, several mechanistic insights have been disclosed by comparing both reaction pathways. The activation barrier for the formation of *aza*-ylide in the **2c**-mediated process (24.7 kcal/mol) was slightly higher than the one in PPh_3_-mediated processes (21.7 kcal/mol). However, the free energy barrier for the decomposition of *aza*-ylide **Int-2** (18.8 kcal/mol) was significantly lower than the one for the decomposition of *aza*-ylide **Int-2′** (33.4 kcal/mol). Notably, the difference in the enthalpy barriers between both processes was much smaller (18.5 kcal/mol vs. 21.1 kcal/mol). Apparently, the proximal SO_2_NH_2_ group in **2c** plays an important role in the protonation of *aza*-ylide **Int-2**. Its significant contribution to entropy greatly lowers the free energy barrier. We also note that the PPh_3_-mediated reduction is more thermodynamically favorable than the **2c**-mediated Staudinger reduction. Therefore, it is most likely that the **2c**-mediated process is largely controlled by favorable kinetics. The rate acceleration in the **2c**-mediated reduction probably originated from the rapid intramolecular proton transfer in the *aza*-ylide decomposition step.

## 3. Materials and Methods

### 3.1. Reagents and General Methods

All anhydrous solvents except toluene and dichloromethane were obtained from common suppliers and used as received. Toluene and dichloromethane were purified according to standard procedures [24]. Analytical thin-layer chromatography was performed on 20 × 50 mm silica gel 60 GF254 plates (Leyan, China). Visualization was accomplished with UV light, potassium permanganate, or ninhydrin stain followed by heating. Flash column chromatography was performed on 200−300 mesh silica gel (Leyan, China). Unless otherwise stated, all reagents were purchased from commercial sources and used without further purification. All reactions were conducted under an atmosphere of nitrogen in oven-dried glassware unless otherwise noted. 2-Iodo-5-methylbenzenesulfonamide was prepared according to the literature methods [25,26]. Bis(4-methoxyphenyl)phosphane [27,28] and organic azides except **1j**, **1s,** and **1w** were prepared according to the literature methods (for **1a** [29], **1b-i**, **1k-m**, **1o**, **1p**, **1r**, **1t-v** [30], **1n** and **1q** [31]).

High-resolution mass spectra (HRMS) were recorded on a Thermo Fisher Scientific’s Q Exactive UHMR Hybrid Quadrupole-Orbitrap Mass Spectrometer LC/MS (ESI); melting points were obtained with INESA WRS-3 apparatus; GC/MS spectra were recorded using a gas chromatograph mass spectrometer (GCMS-QP2010 SE; Shimadzu Corp.) with the electron impact ionization (EI) mode; ^1^H, ^13^C, ^19^F, and ^31^P NMR spectra were recorded on a Bruker AVANCE III 400 MHz spectrometer at 298 K and referenced to residual protium in the NMR solvent (CHCl_3_, δ 7.26 in ^1^H NMR) and the carbon resonances of the solvent (CDCl_3_, δ 77.16 in ^13^C NMR). Chemical shifts were reported in parts per million (ppm, δ) downfield from tetramethylsilane. NMR peaks are described as singlet (s), doublet (d), triplet (t), multiplet (m), complex (comp), approximate (app), and broad (br).

### 3.2. Synthetic Procedures

#### 3.2.1. Synthesis of 2-(Diarylphosphaneyl)-5-methylbenzenesulfonamide

**General Procedure**: Under a N_2_ atmosphere, 2-iodo-5-methylbenzenesulfonamide (12.0 g, 40.0 mmol, 1.0 equiv), diarylphosphine (48.0 mmol, 1.2 equiv), PdCl_2_(dppf) (296.4 mg, 0.40 mmol, 0.01 equiv), diisopropylethylamine (DIPEA, 10.3 g, 80.0 mmol, 2.0 equiv), and anhydrous *N*,*N*-dimethylformamide (40 mL) were placed into a heavy-walled Schlenk tube. The tube was sealed and stirred at 120 °C in an oil bath for 12 h. After completion, the mixture was diluted with dichloromethane (200 mL) and washed with water (200 mL). The separated organic layer was further washed with water (100 mL), dried over anhydrous sodium sulfate, filtered, and concentrated in vacuo. The crude product was purified by flash column chromatography to afford the pure product.

##### 2-(Diphenylphosphaneyl)-5-methylbenzenesulfonamide (**2b**) [32]

Following the **General Procedure**, compound **2b** was obtained as a white solid (11.4 g, 80%). mp 212–213 °C (dichloromethane). R*_f_* = 0.45 (petroleum ether/ethyl acetate, 2:1 *v*/*v*). ^1^H NMR (400 MHz, CDCl_3_) δ 7.99 (dd, *J* = 4.2, 1.8 Hz, 1H), 7.41–7.31 (comp, 6H), 7.31–7.21 (comp, 5H), 7.11 (dd, *J* = 7.7, 3.8 Hz, 1H), 5.51 (br s, 2H), 2.41 (s, 3H). ^13^C{^1^H} NMR (CDCl_3_, 101 MHz): δ 147.1 (d, *J*_C-P_ = 26.2 Hz), 140.4, 136.7, 135.7 (d, *J*_C-P_ = 5.9 Hz, 2C), 133.7 (d, *J*_C-P_ = 19.4 Hz, 4C), 133.2, 131.9 (d, *J*_C-P_ = 20.3 Hz), 129.2 (2C), 128.8 (d, *J*_C-P_ = 7.1 Hz, 4C), 128.7 (d, *J*_C-P_ = 7.8 Hz), 21.3. ^31^P NMR (162 MHz, CDCl_3_): δ −13.2. HRMS-ESI (*m*/*z*) for C_19_H_18_NO_2_PS [M+H]^+^: calcd 356.0869, found 356.0861.

##### 2-(Bis(4-methoxyphenyl)phosphaneyl)-5-methylbenzenesulfonamide (**2c**)

Following the **General Procedure**, the reaction was conducted on a 2.00-mmol scale with higher loading of palladium catalyst (0.02 equiv) instead. Compound **2c** was obtained as a white solid (423.7 mg, 51%). mp 90–93 °C (dichloromethane). R*_f_* = 0.35 (petroleum ether/ethyl acetate, 2:1 *v*/*v*). ^1^H NMR (400 MHz, CDCl_3_) δ 7.96 (dd, *J* = 3.9, 1.9 Hz, 1H), 7.31–7.23 (m, 1H), 7.19 (app t, *J* = 8.2 Hz, 4H), 7.10 (dd, *J* = 7.8, 3.8 Hz, 1H), 6.88 (app d, *J* = 7.9 Hz, 4H), 5.47 (br s, 2H), 3.80 (s, 6H), 2.40 (s, 3H). ^13^C{^1^H} NMR (CDCl_3_, 101 MHz): δ 160.6 (2C), 146.7 (d, *J*_C-P_ = 25.7 Hz), 140.0, 136.2, 135.2 (d, *J*_C-P_ = 20.7 Hz, 4C), 133.1 (d, *J*_C-P_ = 21.0 Hz), 133.1, 128.7 (d, *J*_C-P_ = 4.4 Hz), 126.6 (d, *J*_C-P_ = 2.6 Hz, 2C), 114.5 (d, *J*_C-P_ = 8.2 Hz, 4C), 55.4 (2C), 21.3. ^31^P NMR (162 MHz, CDCl_3_): δ −15.9. HRMS-ESI (*m*/*z*) for C_21_H_22_NO_4_PS [M+H]^+^: calcd 416.1080, found 416.1074.

#### 3.2.2. Synthesis of Organic Azides

Organic azides except **1j**, **1s,** and **1w** were prepared from the literature methods and their analytical data were consistent with the literature data (see Appendix A). Azides **1j**, **1s** and **1w** were prepared by the procedures below.

##### Perfluorophenyl 4-azidobenzoate (**1j**)

A solution of 4-azidobenzoic acid (163.0 mg, 1.00 mmol, 1.0 equiv) and 2,3,4,5,6-pentafluorophenol (220.8 mg, 1.20 mmol, 1.2 equiv) in dichloromethane (5 mL) was cooled down to 0 °C. To this solution was added 1-ethyl-3-(3-dimethylaminopropyl)carbodiimide hydrochloride (287.6 mg, 1.50 mmol, 1.5 equiv) and 4-dimethylaminopyridine (12.2 mg, 0.10 mmol, 0.1 equiv). The mixture was allowed to warm up to room temperature and stirred for 12 h. The solvent was concentrated in vacuo and the crude product was purified by flash column chromatography to afford the product **1j** as a white solid (255.8 mg, 78%). mp 78–79 °C (dichloromethane). R*_f_* = 0.46 (petroleum ether/ethyl acetate, 100:1 *v*/*v*). ^1^H NMR (400 MHz, CDCl_3_): δ 8.19 (d, *J* = 8.8 Hz, 2H), 7.17 (d, *J* = 8.8 Hz, 2H). ^13^C{^1^H} NMR (CDCl_3_, 101 MHz): δ 161.9, 146.9, 143.0–140.1 (m, 2C), 141.3–138.1 (m), 139.7–136.6 (m, 2C), 132.8 (2C), 125.9–125.2 (m), 123.4, 119.5 (2C). ^19^F NMR (377 MHz, CDCl_3_): δ −152.42 to −152.57 (m, 2F), −157.83 (t, *J* = 21.7 Hz), −162.16 to −162.37 (m, 2F).

##### 2-Azido-*N*-(4-bromobenzyl)acetamide (**1s**)

*N*-(4-bromobenzyl)-2-chloroacetamide (7.88 g, 30.0 mmol, 1.0 equiv), sodium fluoride (3.15 g, 75.0 mmol, 2.5 equiv) and trimethylsilyl azide (5.18 g, 45.0 mmol, 1.5 equiv) were dissolved in THF/water (75 mL, 3:1 *v*:*v*). The mixture was heated at 85 °C for 12 h. The reaction mixture was concentrated in vacuo and extracted with ethyl acetate (3 × 80 mL). The organic layer was separated, washed with saturated brine, dried over anhydrous sodium sulfate, filtered, and concentrated in vacuo. The crude mixture was purified by flash column chromatography to afford the product **1s** as a white solid (6.78 g, 84%). mp 93.0 °C (dichloromethane). R*_f_* = 0.23 (petroleum ether/ethyl acetate, 3:1 *v*/*v*). ^1^H NMR (400 MHz, CDCl_3_): δ 7.47 (d, *J* = 8.4 Hz, 2H), 7.16 (d, *J* = 8.4 Hz, 2H), 6.62 (br, 1H), 4.43 (d, *J* = 6.0 Hz, 2H), 4.05 (s, 2H). ^13^C{^1^H} NMR (CDCl_3_, 101 MHz): δ 166.6, 136.7, 132.1 (2C), 129.7 (2C), 121.9, 52.9, 43.0. HRMS-ESI (*m*/*z*) for C_9_H_11_BrN_2_O [M−H]^-^: calcd 266.9886 (^79^Br), 268.9867 (^81^Br), found 266.9984 (^79^Br), 268.9864 (^81^Br).

##### (1*S*,2*R*,4*S*,5*R*)-2-((*R*)-Azido(quinolin-4-yl)methyl)-5-vinylquinuclidine (**1w**)

Following the literature method [33], cinchonine (2.94 g, 10.0 mmol, 1.0 equiv) and triphenylphosphine (3.26 g, 12.0 mmol, 1.2 equiv) were dissolved in 100 mL anhydrous tetrahydrofuran under a N_2_ atmosphere. The mixture was cooled down to 0 °C and then diisopropyl azodicarboxylate (DIAD, 2.32 mL, 12.0 mmol, 1.2 equiv) and a solution of diphenylphosphoryl azide (DPPA, 2.56 mL, 12.0 mmol, 1.2 equiv) in anhydrous tetrahydrofuran (20 mL) were added dropwise. The mixture was stirred at room temperature for 12 h and at 50 °C for another 2 h. The mixture was concentrated in vacuo and the residue was dissolved in dichloromethane and 10% hydrochloric acid (1:1, 100 mL). The aqueous phase was separated and further washed with dichloromethane (4 × 50 mL). Then the aqueous phase was made alkaline (pH ≥ 9) with an excess of concentrated aqueous ammonia (22–25%) and was extracted with dichloromethane (4 × 50 mL). The combined organic phases were dried over anhydrous sodium sulfate and concentrated. The crude product was purified by flash column chromatography and the title compound **1w** was obtained as a yellowish viscous oil (2.24 g, 70%). R*_f_* = 0.14 (ethyl acetate). ^1^H NMR (400 MHz, CDCl_3_): δ 8.92 (dd, *J* = 4.5, 1.4 Hz, 1H), 8.21 (d, *J* = 8.5 Hz, 1H), 8.16 (d, *J* = 8.4 Hz, 1H), 7.79–7.69 (m, 1H), 7.68–7.55 (m, 1H), 7.38 (dd, *J* = 4.5, 1.3 Hz, 1H), 5.81–5.65 (m, 1H), 5.11 (d, *J* = 10.6 Hz, 1H), 5.03–4.86 (comp, 2H), 3.43–3.11 (comp, 3H), 2.96–2.74 (comp, 2H), 2.32–2.15 (m, 1H), 1.65–1.59 (m, 1H), 1.59–1.46 (comp, 2H), 1.45–1.29 (m, 1H), 0.77–0.63 (m, 1H). ^13^C{^1^H} NMR (CDCl_3_, 101 MHz): δ 150.1, 148.9, 142.4, 141.4, 130.8, 129.6, 127.3, 126.7, 123.1, 120.4, 114.7, 62.7, 59.7, 56.1, 41.0, 39.5, 28.0, 27.3, 26.2. HRMS-ESI (*m*/*z*) for C_19_H_21_N_5_ [M+H]^+^: calcd 320.1870, found 320.1863.

#### 3.2.3. Synthesis of Amines

**General Procedure A (for the substrates 1a–1m)**: Under a N_2_ atmosphere, the corresponding organic azide (1.00 mmol, 1.0 equiv) was dissolved in anhydrous tetrahydrofuran (10 mL), followed by the addition of **2c** (1.10 mmol, 1.1 equiv). The mixture was stirred at 30 °C for 3 h. Upon completion, the mixture was concentrated in vacuo and the crude product was purified by flash column chromatography.

**General Procedure B (for the substrates 1n–1w)**: Under a N_2_ atmosphere, the corresponding organic azide (1.00 mmol, 1.0 equiv) was dissolved in anhydrous tetrahydrofuran (10 mL), followed by the addition of **2c** (1.10 mmol, 1.1 equiv). The mixture was stirred at 60 °C for 15 h. Upon completion, the mixture was concentrated in vacuo and the crude product was purified by flash column chromatography. For **1w**, 5.00 mmol of azide was used instead.

All amines except **3j** and **3s** are known compounds and their analytical data were consistent with the literature data (see Appendix A). Their isolated yields were reported in Figure 2.

##### Perfluorophenyl 4-aminobenzoate (**3j**)

Following **General Procedure A**, the title compound **3j** was obtained as a white solid (275.7 mg, 91%) from the azide **1j** (1.00 mmol, 329.0 mg). mp 129 °C (dichloromethane). R*_f_* = 0.44 (petroleum ether/ethyl acetate, 3:1 *v*/*v*). ^1^H NMR (400 MHz, CDCl_3_): δ 7.99 (d, *J* = 8.7 Hz, 2H), 6.69 (d, *J* = 8.7 Hz, 2H), 4.29 (br, 2H). ^13^C{^1^H} NMR (CDCl_3_, 101 MHz): δ 162.7, 152.6, 143.3–140.1 (m, 2C), 141.0–137.8 (m), 139.7–136.4 (m, 2C), 133.2 (2C), 126.2–125.6 (m), 115.8, 114.0 (2C). ^19^F NMR (377 MHz, CDCl_3_): δ −152.56 to −152.76 (m, 2F), −158.88 (t, *J* = 21.6 Hz), −162.87 (td, *J* = 22.4, 5.0 Hz, 2F). GC-MS (EI, 70 eV) *m*/*z*: 303 [M]^+^.

##### 2-Amino-*N*-(4-bromobenzyl)acetamide (**3s**)

Following **General Procedure B**, the title compound was obtained as a white solid (178.0 mg, 82%) from the azide **1s** (0.50 mmol, 121.5 mg). mp 93 °C (dichloromethane). R*_f_* = 0.19 (ethyl acetate/methanol/aqueous ammonia, 100:10:1 *v*/*v*/*v*). ^1^H NMR (400 MHz, CDCl_3_): δ 7.65 (br, 1H), 7.44 (d, *J* = 8.4 Hz, 2H), 7.16 (d, *J* = 8.4 Hz, 2H), 4.41 (d, *J* = 6.1 Hz, 2H), 3.39 (s, 2H), 1.61 (br, 2H, overlapped water). ^13^C{^1^H} NMR (101 MHz, CDCl_3_): δ 172.8, 137.6, 131.9, 129.6, 121.4, 44.8, 42.5. HRMS-ESI (*m*/*z*) for C_9_H_11_BrN_2_O [M+H]^+^: calcd 243.0128 (^79^Br), 245.0107 (^81^Br), found 243.0124 (^79^Br), 245.0104 (^81^Br).

### 3.3. Computational Methods

The calculations were carried out with the Gaussian 09 software package [34]. The structures were optimized by the density functional theory (DFT) [35] with the B3LYP functional [36,37] with basis set 6-31G(d) [38,39] in the gas phase. Frequency analysis was conducted at the same level of theory to verify the stationary points to be real minima or saddle points and to obtain the thermodynamic energy corrections at 298.15 K. Intrinsic reaction coordinate (IRC) [40,41,42] calculations were performed to confirm the connection between two correct minima for a transition state. More accurate electronic energy results were refined by calculating the single-point energy at the B3LYP-D3(BJ) [43]/6-311++G(2df, 2p) [38,39] level of theory with the SMD model [44] (solvent = THF).

## 4. Conclusions

In conclusion, we developed the first *ortho*-phosphinoarenesulfonamide-mediated Staudinger reduction without the need for water. Computational studies suggest that the *ortho*-SO_2_NH_2_ substituent of the phosphine reagent is significant for favorable reaction kinetics. We are currently investigating other *aza*-ylide-driven Staudinger-type transformations which will be reported in due course.

## Data Availability

Not applicable.

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
