# Peer review of "Ortho-Phosphinoarenesulfonamide-Mediated Staudinger Reduction of Aryl and Alkyl Azides"

_molecules, 2022, doi:10.3390/molecules27175707_

Round 1
Reviewer 1 Report
The authors reported a phosphinoarenesulfonamide-mediated Staudinger reaction to reduce azides to amines under mild conditions in good to excellent yields. The reaction gives a cyclic azaylide instead of phosphine oxide, and therefore no hydrolysis step is needed. Aryl azides work better than alkyl azides. The work included a computation component, and the application to a natural product. The work should be of interest to the general audience of Molecules, and is recommended for publications after changes can be made below.
1. The experimental protocol for the Staudinger reduction of cinchonine-derived azide 1w ïƒ 3w is missing. The characterization spectra are missing as well.
2. Fig. 1 is blurry, difficult to see the details especially the numeric numbers.
3. Dichloromethane was included in the mp data. What was it for?
4. There are a number of typos in the text and the experimental section.
5. Ref. 32 is missing the source.
Author Response
Responses to Reviewer #1:
Question 1. The experimental protocol for the Staudinger reduction of cinchonine-derived azide 1w to 3w is missing. The characterization spectra are missing as well.
Response: Thanks for your comment. The experiment from 1w to 3w followed General Procedure B. We have changed “General Procedure B (for the substrates 1n-1v)” to “General Procedure B (for the substrates 1n-1w)…For 1w, 5.00 mmol of azide was used instead.” in the revised manuscript. The characterization spectra of 1w have already been provided in the original manuscript. The NMR spectra of the known compound 3w (not shown) were consistent with the literature data.
Question 2. Fig. 1 is blurry, difficult to see the details especially the numeric numbers.
Response: Thanks for your comment. We apologize for the inconvenience. The poor quality of Figure 1 may have been caused by the version of ChemDraw that we used. We have now converted all ChemDraw pictures into high-resolution tiff files in the revised manuscript.
Question 3. Dichloromethane was included in the mp data. What was it for?
Response: Thanks for your question. The melting point of a solid is known to be affected by the solvent. Indeed, the solvent (dichloromethane) in the parenthesis is the solvent that was used to form the solid samples for analysis. In our laboratory, we transferred the majority of each solid sample into a vial for storage. The small amount of sample remaining in the flask was redissolved in dichloromethane, transferred to a smaller vial and dried in vacuo for melting point examination.
Question 4. There are a number of typos in the text and the experimental section.
Response: Thanks for your comment. We have gone through the manuscript again and corrected any typos. Moreover, Dr. Christopher M. Plummer has checked the language and grammar for us.
Question 5. Ref. 32 is missing the source.
Response: Thanks for your comment. Ref. 32 is a Chinese patent. We have added the label “Chinese Patent” to clarify it.
Reviewer 2 Report
The manuscript submitted by Li and co-workers describes a novel type of ortho-phosphinoarenesulfonamide reagents for the Stuadinger reaction that allow the reaction to be carried out in the absence of water. This is an interesting manuscript that I would recommend publication after the considerations mentioned below are clarified.
The authors might consider changing the title of the manuscript to try to include the "without the need for water", or something similar, as this is one of the key differences enabled by the reagents developed.
Figure 1 (DFT) has very low resolution (quality) and is very difficult to read. This should be fixed before publication.
For the DFT part, the authors should put their proposed mechanistic pathway with the developed phosphinoarenesulfonamide reagents more in the context with common PPh3, which was also described previously in ref. 23 in the manuscript (J. Org. Chem. 2004, 69, 4299-4308). The discussion around DFT falls short, in my opinion.
I would recommend listing the literature references for the synthesized organic azides 1 in the Experimental section, i.e., for each compound 1, list the literature reference in which it is described - to make it easier for the reader to find the appropriate literature references. I would recommend doing the same for products 3.
Most compounds have already been described in the literature, but the new compounds are well characterized.
For solid compounds, especially for 2b and 2c, newly developed reagents, I would recommend adding the melting point values.
How was the stereochemistry determined (checked) in 1w and 3w? This should be commented in the manuscript.
The use of "comp" as a complex to describe the splitting pattern of the proton resonance in 1H NMR spectra, e.g., for compounds 2b and 1w, is very unusual. I would suggest omitting "comp" and using multiplet (m) instead.
The fate of B-TAP as a side product after the reaction is not clear - can B-TAP be regenerated as a byproduct after the reaction is complete? This would be very interesting from a green chemistry point of view.
Author Response
Responses to Reviewer #2:
Question 1. The authors might consider changing the title of the manuscript to try to include the "without the need for water", or something similar, as this is one of the key differences enabled by the reagents developed.
Response: Thanks for your suggestion. As you suggested, we did comtemplate if we should add some words such as “water-free” or “without the need for water” in the title, but these words could be a little bit “misleading” since the reaction can be run with water as well. Instead, we decided to highlight this feature in the text rather than in the title. In addition. we have one sentence “we developed the first ortho-phosphinoarenesulfonamide-mediated Staudinger reduction without the need for water.” in the conclusion.
Question 2. Figure 1 (DFT) has very low resolution (quality) and is very difficult to read. This should be fixed before publication.
Response: Thanks for your comment. We apologize for the inconvenience. The poor quality of Figure 1 may have been caused by the version of ChemDraw that we used. We have now converted all ChemDraw pictures into high-resolution tiff files in the revised manuscript.
Question 3. For the DFT part, the authors should put their proposed mechanistic pathway with the developed phosphinoarenesulfonamide reagents more in the context with common PPh3, which was also described previously in ref. 23 in the manuscript (J. Org. Chem. 2004, 69, 4299-4308). The discussion around DFT falls short, in my opinion.
Response: Thanks for your comments. We apologized again for the blurry Figure 1. In the revised manuscript, a high-resolution version of Figure 1 has been attached. We have shown all key intermediates and TSs in the proposed mechanism in Figure 1. We hope that the updated Figure 1 will give you a clear view of the proposed mechanism. Compared to the 2004 JOC report (Ref. 23), we used real substrates instead of simplified model molecules in our DFT studies. The results disclose that the main difference between our reduction and PPh3-mediated reduction is the step of aza-ylide decomposition, not the step of aza-ylide formation. To highlight this key difference, we have added the sentence “Rate acceleration in the 2c-mediated Staudinger reduction is mainly originated from the rapid intramolecular proton transfer in the aza-ylide decomposition step.” in the revised manuscript.
Question 4. I would recommend listing the literature references for the synthesized organic azides 1 in the Experimental section, i.e., for each compound 1, list the literature reference in which it is described - to make it easier for the reader to find the appropriate literature references. I would recommend doing the same for products 3.
Response: Thanks for your suggestion. Literature methods used for the preparation of organic azides 1 have now been assigned to specific substrates in the revised manuscript. We modified the original text into “organic azides except 1j, 1s, and 1w were prepared according to the literature methods (for 1a[29], 1b-i, 1k-m, 1o, 1p, 1r, 1t-v[30], 1n and 1q[31]).”. In addition, we provide a full reference list for the characterization data of the known azides 1 and products 3 in the updated SI.
Question 5. For solid compounds, especially for 2b and 2c, newly developed reagents, I would recommend adding the melting point values.
Response: Thanks for your suggestion. We have added the melting point data of 2b and 2c in the revised manuscript.
Question 6. How was the stereochemistry determined (checked) in 1w and 3w? This should be commented in the manuscript.
Response: Thanks for your question. In our report, 1w was prepared by a Mitsunobu reaction from cinchonine, which is a well-known SN2 process. Although 1w is a new compound, other cinchona alkaloid-derived azides have been prepared using the same procedure (see Ref. 33). An inversion of the stereocenter has been previously reported in other cinchona alkaloid-derived azides. In addition, 3w is a known compound. The NMR spectra of our sample were consistent with the literature data. Therefore, we speculated that the stereochemistry of 1w and 3w is as shown in Scheme 2.
Question 7. The use of "comp" as a complex to describe the splitting pattern of the proton resonance in 1H NMR spectra, e.g., for compounds 2b and 1w, is very unusual. I would suggest omitting "comp" and using multiplet (m) instead.
Response: Thanks for your comment. The symbol “comp” represents multiple peaks belonging to different sets of equivalent protons while the symbol “m” represents multiple peaks belonging to one set of equivalent proton(s). Although the symbol “comp” may not be “widely” used, still many researcher used it (see: Trost, B. M. et al. J. Am. Chem. Soc. 2010, 132, 9206-9218; Doyle, M. P. et al. J. Am. Chem. Soc. 2011, 133, 9572-9579). In our opinion, it may give more information to the audience.
Question 8. The fate of B-TAP as a side product after the reaction is not clear - can B-TAP be regenerated as a byproduct after the reaction is complete? This would be very interesting from a green chemistry point of view.
Response: Thanks for your suggestion. B-TAP can be isolated either by column chromatography or by precipitation. Recycling B-TAP by precipitation is feasible, although it may not be quantitative.
Round 2
Reviewer 1 Report
I am satisfied with the revision and the responses provided by the authors. The manuscript is recommended for publication.